# A Comparative Study of Periodontal Health Status between International and Domestic University Students in Japan

**DOI:** 10.3390/ijerph20053866

**Published:** 2023-02-22

**Authors:** Masanobu Abe, Ai Ohsato, Yuko Fujihara, Kazuto Hoshi, Shintaro Yanagimoto

**Affiliations:** 1Division for Health Service Promotion, The University of Tokyo, Tokyo 113-0033, Japan; 2Department of Oral & Maxillofacial Surgery, The University of Tokyo Hospital, Tokyo 113-8655, Japan

**Keywords:** gingival bleeding, bleeding on probing (BOP), pocket depth, oral health, oral hygiene, oral health behavior, oral health consciousness, periodontal disease

## Abstract

Background: In our previous study, international university students showed a significantly higher dental caries morbidity rate than domestic students. On the other hand, the periodontal health status of international university students has not been clarified yet. In this study, we compared the periodontal health status of international and domestic university students in Japan. Methods: We conducted a retrospective review of the clinical data of the university students that visited a dental clinic in the division for health service promotion at a university in Tokyo for screening between April 2017 and March 2019. Bleeding on probing (BOP), calculus deposition and probing pocket depth (PPD) were investigated. Results: The records of 231 university students (79 international and 152 domestic university students) were analyzed; 84.8% of international students were from Asian countries (*n* = 67). The international university students showed a higher percentage of BOP than domestic students (49.4% and 34.2%, respectively: *p* < 0.05) and they showed more extensive calculus deposition (calculus grading score [CGS]) than domestic university students (1.68 and 1.43, respectively: *p* < 0.01), despite no significant difference in PPD. Conclusions: The current study shows that international university students have poorer periodontal health than domestic students in Japan, even though the result might include many uncertainties and possible biases. To prevent severe periodontitis in the future, regular checkups and thorough oral health care are essential for the university students, especially those from foreign countries.

## 1. Introduction

Periodontal diseases, along with dental caries, are an important public health problem in terms of their high prevalence, affecting approximately 90% of the world’s population [1,2,3]. Periodontal disease is classified into gingivitis and periodontitis. Gingivitis, the mildest form of periodontal disease, is caused by a bacterial biofilm that accumulates on teeth adjacent to the gingiva. Gingivitis does not affect the supporting structures of the teeth and is reversible. On the other hand, periodontitis, the advanced stage of periodontal disease, causes loss of connective tissue and bone support and is the leading cause of tooth loss in adults. In addition to pathogenic microorganisms in biofilms, genetic and environmental factors such as smoking are known to contribute to the cause of these diseases [3]. 

Severe periodontitis is reported to have the sixth highest prevalence in the world (11%) [4]. Although dental caries used to be the leading cause of tooth loss in Japan [5,6], now periodontal disease has overtaken dental caries as the leading cause of tooth loss. Specifically, 30.2% of men and 29.0% of women lose their teeth due to dental caries, and 40.4% of men and 34.9% of women lose their teeth due to periodontal disease [7]. Unlike dental caries, periodontal disease often does not cause severe pain, thus regular checkups by dentists are essential. Furthermore, it has been shown that periodontal disease not only causes tooth loss, but also affects overall health [8]. Various diseases, including respiratory diseases [9,10], cardiovascular diseases [11,12,13], rheumatoid arthritis [14], diabetes [15], and others [16,17,18,19], have been reported to be associated with periodontal disease. Although the periodontal disease is more prevalent in middle-aged and older adult, more than one-third of university students aged less than 20 years are already aware of gum bleeding which is known as a major symptom of periodontal disease. Importantly, the gum bleeding is found to be closely associated with common systemic disorders in late adolescence such as asthma [20,21,22]. The importance of periodontal disease prevention from young age has been increasing [2].

According to the Japan Student Services Organization (JASSO: https://www.jasso.go.jp, accessed on 10 January 2023), the number of foreign students at Japanese institutions of higher education and Japanese language education is currently on the rise, although it is temporarily declining due to the COVID-19 pandemic. In a study by Ohsato et al. [23], authors analyzed the medical records of 554 subjects (138 international and 416 domestic university students) and found no significant difference in dental treatment history between international and domestic university students (49.3% and 48.8%, respectively). However, the incidence of dental caries was significantly higher in international university students than in domestic university students (60.1% and 49.0%, respectively). The indices of decayed, missing, and filled teeth (DMFT) were also significantly higher in international university students than in domestic university students (5.0 and 4.0, respectively). International university students were found to have a significantly higher dental caries morbidity rate than domestic students in Japan [23]. On the other hand, the differences in periodontal health between international and domestic university students are not yet well defined. In this study, the periodontal health status of international and domestic university students was compared.

## 2. Methods

### 2.1. Study Design and Population

Clinical data of university students who visited a dental clinic at The University of Tokyo for screening purposes (not for symptomatic or dental treatment purposes) between April 2017 and March 2019 were retrospectively analyzed. Students who held Japanese nationality or who were permanent residents of Japan were classified into the group of domestic university students in Japan. Of the 374 university students who visited the dental clinic for initial dental checkups, the records of 231 university students under 25 years of age (including 79 international students) were included in the analysis. No specific undergraduate or graduate school students were targeted.

Periodontal health status was determined by three dentists’ individual examinations on separate students for three parameters: probing pocket depth (PPD), bleeding on probing (BOP), and calculus grading scale (CGS) in the fully erupted permanent dentition excluding wisdom teeth. One identical dental hygienist was present during all examinations to ensure that the examinations were performed properly. A community periodontal index (CPI) probe (YDM, Tokyo, Japan) was used to measure each tooth at six sites (mesiobuccal, mid-buccal, distobuccal, distolingual, mid-lingual, and mesiolingual) for the evaluation of PPD and BOP [24]. The PPD value of each tooth was determined as the deepest of the six locations listed above. The PPD value of each student was determined as the mean value of the PPD of each tooth. For BOP, if bleeding was observed in even one location, the student was classified as having BOP. CGS was determined as follows: NONE: no calculus deposition (scored as 1), MILD: calculus deposition on less than one-half of the tooth surface (scored as 2), SEVERE: calculus deposition on more than one-half of the tooth surface and/or extending below the gingival margin (scored as 3) [23]. 

This study was approved by the Research Ethics Committee of the University of Tokyo (approval number 13–146): “Retrospective analyses of medical and health record information retained by the division for health service promotion, the University of Tokyo.” 

### 2.2. Statistical Analyses

Statistical analysis was performed using the χ2 test for BOP evaluation and the student’s t-test for PPD and CGS evaluation. A value of *p* < 0.05 (two-sided) was accepted as statistically significant. All the analyses were conducted using the statistical software program: Statistical Package for Social Sciences (SPSS version 21.0, IBM Corporation, Armonk, NY, USA). No statistical sample size calculations were conducted.

## 3. Results

### 3.1. Region of Origin of International University Students

The records of all university students under 25 years of age who visited a dental clinic for checkups (not for symptomatic or dental treatment) between April 2017 and March 2019 were analyzed. Of the total 231 university students, 152 were domestic students and 79 were international students. Of the international students, 84.8% were from Asian countries (*n* = 67), which was the highest percentage, followed by North America and Europe, both 5.1% (*n* = 4). In Asia, China accounted for 83.6% (*n* = 56) of all Asian international university students, followed by South Korea with 6.0% (*n* = 4), Singapore and Thailand both with 3.0% (*n* = 2), and Hong Kong, Taiwan, and Malaysia with 1.5% (*n* = 1). Among international university students, 45.6% (*n* = 36) were male and 54.4% (*n* = 43) were female; among domestic university students, 77.6% (*n* = 118) were male, and 22.4% (*n* = 34) were female (Figure 1).

### 3.2. Difference in Bleeding on Probing (BOP) between International and Domestic University Students in Japan

The mean number of remaining teeth for all university students was 27.6 (maximum number of teeth: 28 excluding wisdom teeth). The mean number of remaining teeth for international students was 27.2, while that for domestic students was 27.7. The periodontal status of those remaining teeth was evaluated in this study. Overall, 91 of 231 (39.4%) university students showed BOP. By gender, 60 of 154 (39.0%) males and 31 of 77 (40.3%) females had BOP. There were no significant differences between males and females. 39 of 79 (49.4%) international university students and 52 of 152 (34.2%) domestic university students showed BOP. The international university students showed a higher percentage of BOP than domestic university students (*p* < 0.05). Among international university students, females tended to exhibit BOP at a higher rate than males (55.8% and 41.7%, respectively: *p* = 0.21). On the other hand, among domestic university students, males showed BOP at a higher rate than females (38.1% and 20.6%, respectively: *p* = 0.057), although the difference was not significant (Figure 2, Appendix A). 

### 3.3. Differences in Calculus Deposition between International and Domestic University Students

The mean calculus grading score (CGS) of the total (*n* = 231) was 1.52. By gender, the mean CGS for males (*n* = 154) was 1.55, and for females (*n* = 77) was 1.45, showing no significant difference. The mean CGS of international university students (*n* = 79) was 1.68 and that of domestic university students (*n* = 152) was 1.43. The international university students showed more extensive calculus deposition than domestic students (*p* < 0.01). Among international university students, males tended to have higher CGS than females (1.83 and 1.56, respectively: *p* = 0.10). Similarly, among domestic university students, males tended to have higher CGS than females (1.46 and 1.32, respectively: *p* = 0.24) (Figure 3, Appendix A).

### 3.4. Difference in Probing Pocket Depth (PPD) Status between International and Domestic University Students

The mean PPD of the total university students (*n* = 231) was 1.68 mm. By gender, the mean PPD for males (*n* = 154) was 1.67 mm and for females (*n* = 77) was 1.70 mm, showing no significant difference. The mean PPD of international university students (*n* = 79) was 1.77 mm and that of domestic university students (*n* = 152) was 1.64 mm, showing no significant difference between the two groups. There was no gender difference in PPD for either international university students or domestic university students (Figure 4, Appendix A).

### 3.5. The Association between BOP and PPD in International and Domestic University Students

The mean PPD of the university students with BOP was 1.98 mm, and the mean PPD of the students without BOP was 1.49 mm, showing a large difference (*p* < 0.001). For international university students, the mean PPD for students with BOP was 2.05 mm, while the mean PPD for students without BOP was 1.49 mm. The mean PPD of international students with BOP was significantly larger than that of international students without BOP (*p* < 0.001). The mean PPD for domestic university students with BOP was 1.93 mm, and the mean PPD for domestic university students without BOP was 1.48 mm. The mean PPD of domestic students with BOP was significantly larger than that of domestic students without BOP (*p* < 0.001). Within the population that showed BOP, there was no significant difference in PPD between international and domestic students. Even among the population without BOP, there was no significant difference in PPD between international and domestic students (Figure 5, Appendix A).

## 4. Discussion

The current study showed that international university students in Japan had a higher rate of bleeding on probing (BOP) and more extensive calculus deposition than domestic university students. Although probing pocket depth (PPD) was found to be at a physiological level for both international and domestic students, and no differences were observed, students with BOP showed significantly larger PPD values than those without BOP, regardless of international or domestic students. 

Oral diseases such as caries, periodontal disease, tooth loss, oral infections, oral cancer, and malocclusion are among the most prevalent diseases worldwide and carry serious health and economic burdens that significantly reduce the quality of life of those affected, and their impact is immeasurable [25]. Oral diseases, like most non-communicable diseases (NCDs), are chronic and susceptible to social context, such as economic status. Chronic untreated oral diseases often have serious consequences, not only in terms of pain and other painful symptoms and progression to systemic diseases (e.g., sepsis), but also in terms of reduced quality of life and work productivity. The cost of treating oral diseases also imposes a significant financial burden on households and health care systems [25]. Unfortunately, oral diseases have not been given much importance in global health policy, including in Japan, despite the fact that they are a global public health problem. In recent years, however, the need to treat oral diseases as an urgent priority for global health has begun to be stated [2,25,26,27,28]. Among oral diseases, periodontal disease is of particular public health importance because it occurs with such high frequency that it is estimated to affect 90% of the world’s population [3]. 

Periodontal disease is the most common disease affecting tooth-supporting structures and is therefore a common cause of tooth loss [11,29,30,31]. In Japan, periodontal disease has replaced dental caries as the leading cause of tooth loss [7]. Importantly, periodontal disease has also been shown to be associated with a variety of systemic diseases including respiratory diseases [9,10], cardiovascular diseases [11,12,13], rheumatoid arthritis [14], diabetes [15], and a lot of other disorders [16,17,18,19,32]. Therefore, the importance of prevention and treatment of periodontal disease has been recognized by society and has become a focus of public health in recent years [25,28,33]. The relationship between systemic diseases and periodontal disease has been discussed mainly in middle-aged and older adults. Recently, however, it has been shown that late adolescents who have gingival bleeding are significantly more likely to suffer from systemic diseases such as asthma, otitis media/externa [21]. Based on the above, it is not surprising that the relationship between periodontal disease and lifestyle-related diseases such as diabetes and stroke, which are common in middle-aged and older adults, has already begun during late adolescence.

Although subjective symptoms of periodontal disease usually become apparent after the age of 40s, it is common for young people to develop gingivitis, an early stage of periodontal disease, and in a survey of 17- to 19-year-old university students, 36.5% of them complained of gingival bleeding [21]. This result suggests that one out of every three persons in their late teens already has gingivitis. In addition, Dental Health Division of Health Policy Bureau Ministry of Health in Japan reported that periodontal pockets rapidly become deeper after the age of 20 years [20]. This suggests that periodontal disease has already begun in late adolescence, indicating the need for periodontal disease countermeasures for young people [22].

According to the Japan Student Services Organization (JASSO: https://www.jasso.go.jp accessed on 10 January 2023), an independent administrative agency under the jurisdiction of the Ministry of Education, Culture, Sports, Science and Technology, the number of foreign students at Japanese institutions of higher education and Japanese language education is on the rise (although the situation is currently exceptional due to the COVID-19 pandemic), with the largest number of students from Asian countries. In this study, foreign students from Asia accounted for the largest proportion of foreign students (84.8%), with students from China accounting for a particularly large share (83.6%) of all foreign students from Asia. Therefore, the exclusion of international students from North America and Europe (together comprising 5.1% of all international students) did not significantly change the results of the current study. It is not clear whether the percentage of foreign students from Europe and North America will increase in the future but, at present, foreign students from Asia are by far the largest group of foreign students in Japan. Therefore, it is necessary to consider many social factors such as differences in culture, customs, insurance systems, and medical services in order to provide better oral healthcare services to international students from Asian countries. Since there are still few studies comparing the oral health status of international students and domestic students in Japan, more data needs to be accumulated in the future [23,34]. 

In the current study, 34.2% of domestic university students showed BOP, more than a third of the subjects, which was similar to our previous survey [21]. On the other hand, international university students showed a higher percentage of BOP (49.4%) than domestic university students. A large difference was observed between international and domestic university students in BOP (Figure 2). Among international university students, women tended to have a higher percentage of BOP than men, while among domestic university students, male tended to have a higher percentage of BOP than female, consistent with the result of a previous survey [1]. Although this difference needs to be examined with a larger number of subjects, it is interesting because cultural differences and social backgrounds may be involved. In a study by Ohsato et al. [23], severe calculus deposition was observed in international university students (51.9%) compared with domestic students (31.7%) in Japan [23]. Similar results were obtained in the present study, with international university students showing more extensive calculus deposition than domestic students (*p* < 0.01) (Figure 3). This difference can be attributed to differences in food culture, socioeconomic differences, and lifestyle habits such as brushing teeth and dental visits. University students with BOP had greater PPD values than those without BOP, although, even for students with BOP, the depth of the gingival sulcus was at the physiological level: gingival pocket (Figure 5). The presence of BOP without periodontal pockets indicates the presence of gingivitis. Gingivitis is in a reversible stage in which healthy periodontal tissue can be restored [3,25,35,36,37]. Thus, some measures are needed to prevent the transition from gingivitis to periodontitis.

What measures should be taken to prevent periodontitis in young people? Recently, it has been shown that the frequency and duration of tooth brushing affect gingival health in late adolescence [1,38]. In a survey of 9098 university students aged 17–19 years, regarding the frequency of tooth brushing, it was reported that the risk of gingival bleeding for university students who brush their teeth “less than once” is 2.36 times that of those who brush their teeth “three or more times,” and even for those who brush their teeth “twice” the risk of gingival bleeding is 1.45 times that of those who brush “three or more times.” Regarding the duration of tooth brushing, it is known that university students who brush their teeth “1 minute or less” have 1.57 times the risk of gingival bleeding compared to those who brush “4 minutes or more” and those who brush “2 to 3 minutes” have 1.26 times the risk compared to those who brush “4 minutes or more.” University students who brush their teeth less frequently and for less time have a higher risk of gingival bleeding. This result implies that the risk of periodontal disease decreases as the frequency and duration of tooth brushing increases. Therefore, in addition to dental checkups, it is important to raise oral hygiene and oral health awareness among the younger generation. However, unfortunately, the working-age population from high school graduation (age 18) to age 40 does not have opportunities to receive dental examinations or oral care instruction, except for special examinations limited to targeted occupations in Japan. Considering that periodontal disease begins in late adolescence and becomes apparent in the forties, it seems essential to establish a seamless oral hygiene management system that compensates for this gap period in Japan [2].

The current study revealed that international university students in Japan have poorer periodontal health status than domestic university students. Although the number of university students included in the study was not large and more studies with a larger number of students are needed, the results suggest that regular checkups and thorough oral care are essential for university students, especially international students, in order to prevent periodontitis. It has been suggested that the relationship between periodontal disease and systemic health status already occurs in late adolescence [21,39,40,41]. From the viewpoint of preventing systemic diseases, oral health care in late adolescence will become increasingly important in the future.

## 5. Conclusions

International university students in Japan showed higher percentage of bleeding on probing (BOP) and more extensive calculus deposition than domestic university students, despite no significant difference in probing pocket depth (PPD). Students with BOP have significantly greater PPD values than those without BOP in both international and domestic students, although the values were at physiological levels. To prevent periodontitis, we have to pay more attention to the periodontal health care of university students, especially international students.

## Figures and Tables

**Figure 1 ijerph-20-03866-f001:**
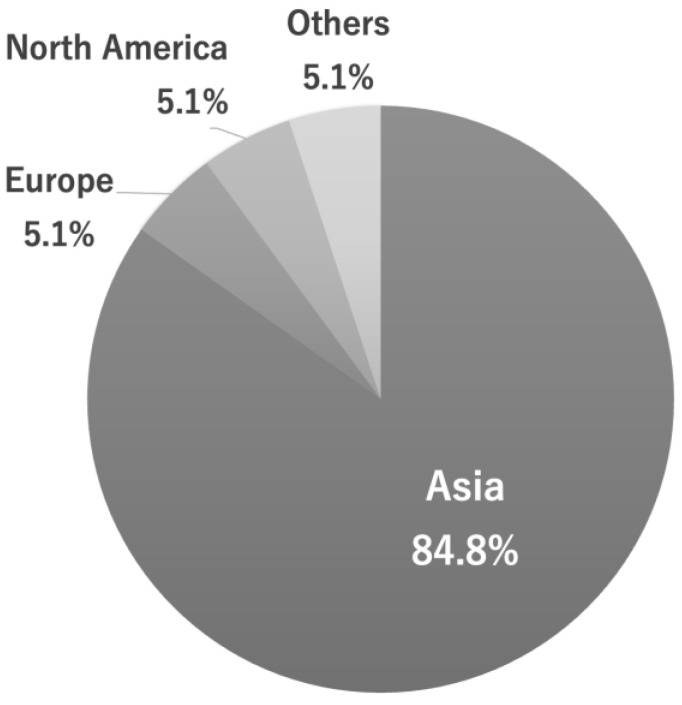
Region of origin of international university students in Japan. Note: Of the 231 university students under the age of 25, 79 were international students. Of these, 84.8% (*n* = 67) were from Asian countries, the highest percentage, followed by 5.1% (*n* = 4) from both North America and Europe.

**Figure 2 ijerph-20-03866-f002:**
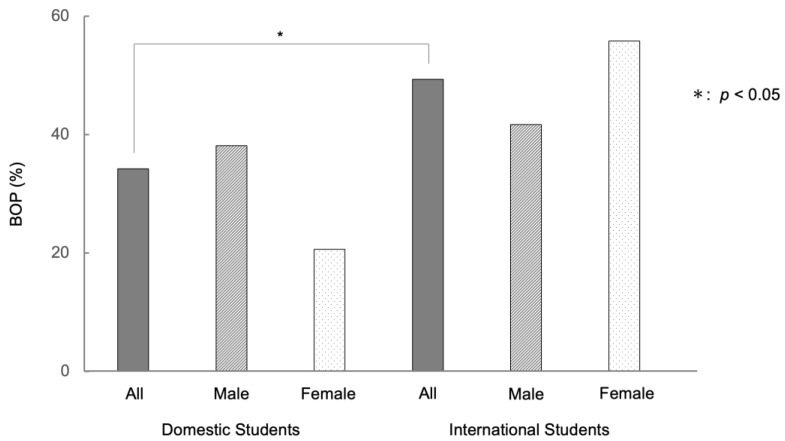
Differences of bleeding on probing (BOP) between international and domestic university students in Japan. Note: The international students showed higher percentage of BOP than domestic students (49.4% and 34.2%, respectively: *p* < 0.05).

**Figure 3 ijerph-20-03866-f003:**
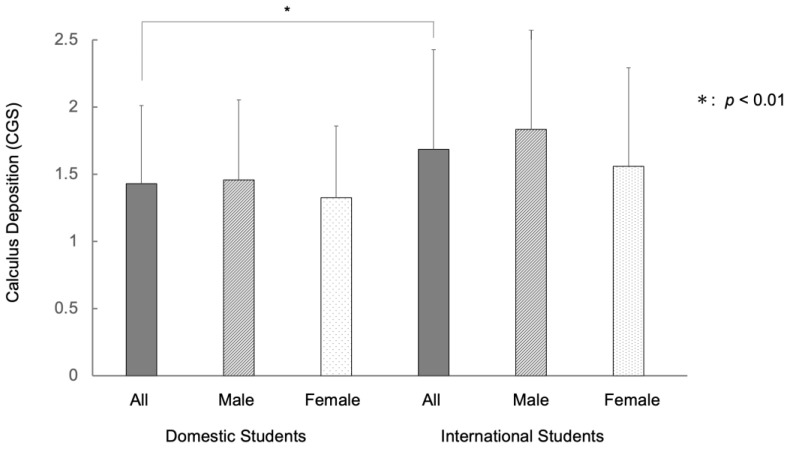
Differences of calculus deposition between international and domestic university students in Japan. Note: The mean CGS of international students (*n* = 79) was 1.68 and that of domestic students (*n* = 152) was 1.43. The international students showed more extensive calculus deposition (*p* < 0.01). Error bars denote the standard deviations of the data.

**Figure 4 ijerph-20-03866-f004:**
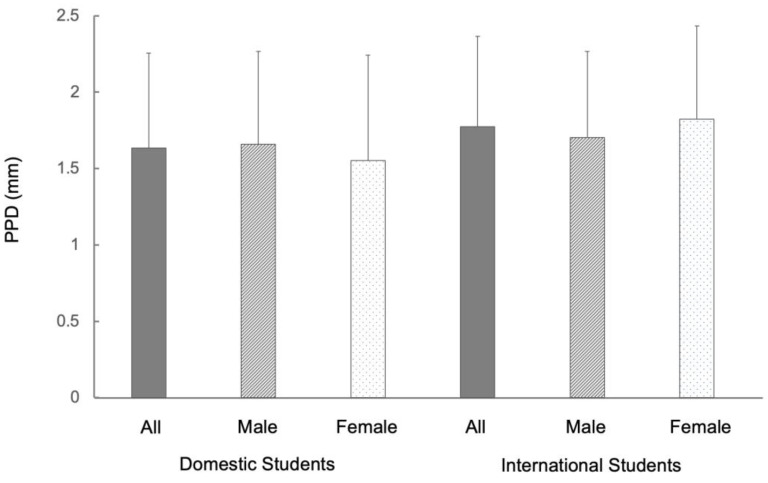
Differences in probing pocket depth (PPD) status between international and domestic students in Japan. Note: The mean PPD of international students (*n* = 79) was 1.77 mm and that of domestic students (*n* = 152) was 1.64 mm, showing no significant difference between the two groups. Error bars denote the standard deviations of the data.

**Figure 5 ijerph-20-03866-f005:**
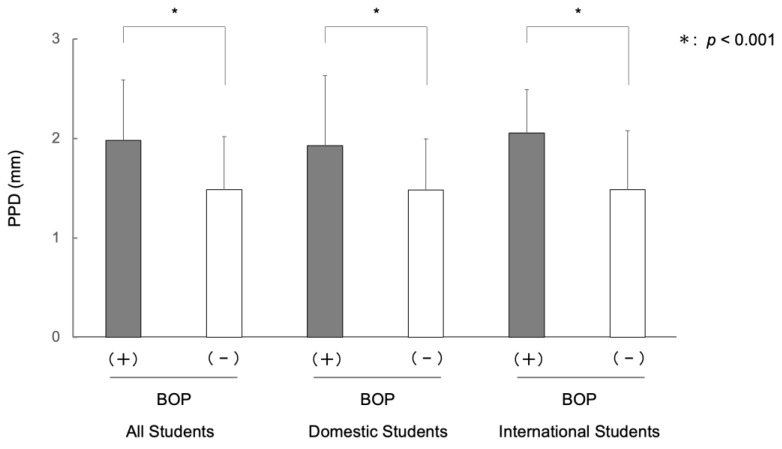
The association of bleeding on probing (BOP) with probing pocket depth (PPD) in international and domestic university students in Japan. Note: The mean PPD for students with BOP was significantly greater than that for students without BOP (*p* < 0.001). For both international and domestic students, the mean PPD for students with BOP was greater than that for students without BOP (*p* < 0.001); there was no significant difference in PPD between international and domestic students for both students with BOP and students without BOP. Error bars denote the standard deviations of the data.

## Data Availability

Not applicable.

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
