# Peer review of "A Comparative Study of Periodontal Health Status between International and Domestic University Students in Japan"

_ijerph, 2023, doi:10.3390/ijerph20053866_

Round 1

Reviewer 1 Report

Dear Author, 

Please view the comments attached 

Regards

Author Response

We appreciate the reviewer's precise and helpful comments.

Introduction

  1. It is mentioned that- According to the Japan Student Services Organization (JASSO), the number of foreign students at Japanese institutions of higher education and Japanese language education is currently on the rise, although it is temporarily declining due to the COVID-19 pandemic…. Kindly support with source and year

Answer

The official website of Japan Student Services Organization (JASSO) was indicated in Introduction section.

  1. It is mentioned that- In a previous study, we analyzed the medical kindly revise as- authors analyzed…. Instead

Answer

We have corrected the point the reviewer indicated.

  1. Kindly mention the detailed aim of the study at the end of the introduction

Answer

Aim of the study was added at the end of the section.

  1. Kindly consider whether the first two paragraphs can be interchanged in order of narration with periodontitis as the beginning of the introduction

Answer

First two paragraphs were interchanged.

Methodology

  1. Dear authors, kindly be specific with the reference of university students- kindly comment whether any group specialty was considered or in general any major students graduating irrespective of the field was considered

Answer

Students from all faculties were included in the analysis. This was added into the section.

  1. Kindly mention the sample size

Answer

Thank you so much for this critical comment. Actually, no statistical sample size calculations were conducted in this retrospective study and this was mentioned in the Methods section. However, as of May 2017, international students accounted for 11.9% of the University of Tokyo's student population. Assuming a maximum permissible error of 5 and a confidence level of 95%, the sample size is to be 161, and the current analysis is considered acceptable.

  1. Kindly mention the university the participants visited for screening

Answer

The name of the University (The University of Tokyo) was added.

  1. Kindly comment on institutional approval for the study

Answer

Institutional approval was clearly described in Methodology section.

  1. Methodology can be further kindly consider- kindly mention the section criteria in detail

Answer

Additions have been made to the methodology.

Results

  1. Kindly mention the no: of domestic students considered too in the beginning of the results

Answer

No of domestic students was also mentioned in the beginning of the Results section.

  1. Kindly mention the inference of the study at the end of the results

Answer

On the advice of this reviewer and the advice of other reviewers, we have decided to discuss the new results of this study at the beginning of the Discussion section.

Discussion

  1. Kindly refer as present/ current study instead of we/ our in the narration. Kindly revise wherever applicable

Answer

Through the manuscript, we have used present/current study instead of we/our.

  1. It is mentioned that-was significantly higher in foreign university students than in domestic university students in Japan; ). Among international university students, women … kindly revise as foreign/ international students instead.. kindly revise wherever applicable

Answer

Thank you for this reviewers' detailed remarks. 'Foreign university student' was revised to 'international university student'.

  1. Dear authors, kindly note that the first paragraph in the discussion is already addressed in detail in the introduction. kindly revise

Answer

On the advice of the reviewers, the section was substantially revised.

  1. Repetitions are note in the kindly verify

Answer

Repetitions were removed.

  1. Kindly cite the authors while discussing their studies/ comparing the results with the present study

Answer

We cited the authors whenever possible.

  1. Kindly comment on below
    1. In a previous study, severe calculus deposition was observed in international university students (51.9%) compared with domestic students (31.7%) in Japan [1].
    2. On the other hands, no comparison had been made between international and domestic university students with regard to periodontal health status

Answer

Thank you for your accurate point of view. This sentence has been removed because of the inconsistency.

Tables/ graphs/ figures

  1. Kindly include the results / data representation in appropriate forms at the end of the manuscript. Though figures are cited are not included. kindly verify.

Answer

Figures were inserted. We are so sorry about that.

Reviewer 2 Report

This article entitled “A Comparative Study of Periodontal Health Status between 2 International and Domestic University Students in Japan” aims to compare the state of periodontal health between foreign students and Japanese students at the University of Japan.

Introduction

It would be advisable to analyze periodontal diseases according to their type and not as a group due to their great variability, incidence and consequences of their severity.

Material and method

- All the patients were analyzed by the three operators individually or the operators analyzed the group as a whole.

- In the event that each operator analyzes a certain number of students. Were your criteria properly calibrated? some possible variability was taken into account in the results due to the fact that different operators were involved?

As the authors comment, the permanent dentition was taken into account except for third molars. In the selected patients, was there any dental loss? in which groups? did it influence the results?

or on the contrary, 100% of the patients presented all the teeth

Results

- When analyzing the results, were socioeconomic or cultural differences between the groups taken into account?

 the authors comment that the frequency and quality of brushing and oral hygiene of the participants were analyzed. in addition to giving them instructions to improve it. Were there changes in the improvement and frequency of brushing during the study? Did the combination of improved hygiene and the presence of caries in any of the groups have any influence on the results?

Could there have been any differences in the results due to the sex of the patients?

Figures could not be parsed because they are not available in the submitted text

Discussion and conclusions

We recommend authors to review this section. The first paragraphs should be found in the first section and not within the discussion since they do not evaluate the results or compare them with other studies.

We also recommend re-evaluating the data provided and focusing on analyzing the results of the study being if possible compared with other studies

It would be interesting to analyze the motivation for carrying out this study and the importance of the results, as well as whether it is possible to extrapolate it to other universities.

Author Response

We appreciate the reviewers' kind and constructive comments, including suggestions for future research plans.

Introduction

  1. It would be advisable to analyze periodontal diseases according to their type and not as a group due to their great variability, incidence and consequences of their severity.

Answer

As the reviewer stated, it is desirable to analyze periodontal disease by type. However, it is difficult to classify types of periodontal disease in adolescents because they do not have deep periodontal pockets.

Material and method

  1. All the patients were analyzed by the three operators individually or the operators analyzed the group as a whole.

Answer

Thank you so much for this critical comment. Actually, three operators performed individual periodontal examinations on separate students. This information was added into the Methods section.

  1. In the event that each operator analyzes a certain number of students. Were your criteria properly calibrated? some possible variability was taken into account in the results due to the fact that different operators were involved?

Answer

The dentists had a preliminary discussion regarding examination procedures and evaluation. In addition, one identical dental hygienist was present during all examinations to ensure that the examinations were performed properly. This information was added into the Methods section.

  1. As the authors comment, the permanent dentition was taken into account except for third molars. In the selected patients, was there any dental loss? in which groups? did it influence the results? or on the contrary, 100% of the patients presented all the teeth

Answer

We thank the reviewer for his important remarks. The number of remaining teeth for international students was 27.2 and that for domestic students was 27.7 excluding wisdom teeth (maximum number of teeth: 28). This information was added into the Result section. However, the cause of tooth loss is unknown and its impact on the results of this study is not clear. We will focus on this in future studies. 

Results

  1. When analyzing the results, were socioeconomic or cultural differences between the groups taken into account?

Answer

As the reviewer pointed out, socioeconomic and cultural differences between groups are very important, although they were not pursued in this study. We have described about this issue in the Discussion section and would like to address in the future study.

  1. the authors comment that the frequency and quality of brushing and oral hygiene of the participants were analyzed. in addition to giving them instructions to improve it. Were there changes in the improvement and frequency of brushing during the study? Did the combination of improved hygiene and the presence of caries in any of the groups have any influence on the results?

Answer

We conducted a survey study on the frequency and duration of subjects' brushing in a previous report, but did not do so in this study. Although we provided instruction to improve oral health, we do not have sufficient data on improvements in brushing and changes in frequency, making it difficult to report on these findings in this study.

  1. Could there have been any differences in the results due to the sex of the patients?

Answer

Among international students, females tended to show a higher percentage of BOP than males, although the difference was not significant. Among domestic university students, males were more likely than females to exhibit BOP, although there were no significant differences. No gender differences were found in the results of the other analyses.

  1. Figures could not be parsed because they are not available in the submitted text

Answer

Figures were inserted. We are so sorry about that.

Discussion and conclusions

  1. We recommend authors to review this section. The first paragraphs should be found in the first section and not within the discussion since they do not evaluate the results or compare them with other studies.

Answer

Thank you very much. We reviewed this section and revised.

  1. We also recommend re-evaluating the data provided and focusing on analyzing the results of the study being if possible compared with other studies

It would be interesting to analyze the motivation for carrying out this study and the importance of the results, as well as whether it is possible to extrapolate it to other universities.

Answer

We too are interested in doing a comparative analysis of the study results in comparison to other studies. Since there are no similar studies at this time, we would like to plan a joint study with another university as suggested by the reviewer.

Reviewer 3 Report

Thank you for allowing me to review the manuscript entitled "A Comparative Study of Periodontal Health Status between International and Domestic University Students in Japan". The article is interesting, but it is necessary to correct some parts, and the realization itself needs to be revised.

1. Abstract - is long and requires extensive revision. Please shorten the objectives, and state only the purpose of the work. The first sentence of the result is a repetition of the goal of the manuscript - please remove it. Reduce the results to three sentences of the most important results.

2. In a more scientific manuscript, it is better to use current, present or this study than we/our study

3. What were the inclusion and exclusion criteria?

4. Who performed the measurements and how?

5. How are the data presented? Was the normality of the distribution of the obtained data measured? How?

6. How was the sample size measured? For example, if you have a large number of international students, 231 is insignificant.

7. The obtained results are stated only in words. Please present them in a table so that the difference between the examined groups can be seen.

8. Figures are mentioned in the text which is not visible.

9. The discussion started with the earlier results of different author studies; please start with the new results of this study.

10. What are the limitations of the study?

11. Reference are not according Journal style.

Author Response

We appreciate the reviewer's accurate and constructive comments. Thank you very much.

  1. Abstract - is long and requires extensive revision. Please shorten the objectives, and state only the purpose of the work. The first sentence of the result is a repetition of the goal of the manuscript - please remove it. Reduce the results to three sentences of the most important results.

Answer

Regarding the abstract, the objectives has been revised. In addition, the number of sentences of the results was reduced.

  1. In a more scientific manuscript, it is better to use current, present or this study than we/our study.

Answer

Through the manuscript, we have revised this point.

  1. What were the inclusion and exclusion criteria?

Answer

Thank you very much. Owing to this comment, we revised Methods section.

  1. Who performed the measurements and how?

Answer

Three operators performed individual periodontal examinations on separate students. This information was added into the Methods section.

  1. How are the data presented? Was the normality of the distribution of the obtained data measured? How?

Answer

We are sorry for the lack of information. We have prepared graphs and tables of results. The values are shown as the mean of the obtained data (+/- standard deviation).

  1. How was the sample size measured? For example, if you have a large number of international students, 231 is

Answer

Thank you so much for this critical comment. Actually, no statistical sample size calculations were conducted in this retrospective study. However, as of May 2017, international students accounted for 11.9% of the University of Tokyo's student population. Assuming a maximum permissible error of 5 and a confidence level of 95%, the sample size is to be 161, and the current analysis is considered acceptable.

  1. The obtained results are stated only in words. Please present them in a table so that the difference between the examined groups can be seen.

Answer

Thank you for this kind advice. We prepared a Table (Supplementary Table 1).

  1. Figures are mentioned in the text which is not

Answer

We are so sorry. Figures have been inserted.

  1. The discussion started with the earlier results of different author studies; please start with the new results of this study.

Answer

On the advice of the reviewer, the new results of this study was discussed at the beginning of the Discussion section.

  1. What are the limitations of the study?

Answer

Limitations were described in Abstract and Discussion section.

  1. Reference are not according Journal style.

Answer

The format of the reference has been adapted to the style of this journal.

Round 2

Reviewer 2 Report

Thank you very much for reconsidering all the recommended points. Honestly, I think the article presents a better description and is more understandable by the reader.

Reviewer 3 Report

I do not have any more comments. The paper is simple and on a small sample of respondents, but solidly written.